# Conventional Cancer Therapies Can Accelerate Malignant Potential of Cancer Cells by Activating Cancer-Associated Fibroblasts in Esophageal Cancer Models

**DOI:** 10.3390/cancers15112971

**Published:** 2023-05-30

**Authors:** Satoshi Komoto, Kazuhiro Noma, Takuya Kato, Teruki Kobayashi, Noriyuki Nishiwaki, Toru Narusaka, Hiroaki Sato, Yuki Katsura, Hajime Kashima, Satoru Kikuchi, Toshiaki Ohara, Hiroshi Tazawa, Toshiyoshi Fujiwara

**Affiliations:** 1Department of Gastroenterological Surgery, Okayama University Graduate School of Medicine, Dentistry, and Pharmaceutical Sciences, Okayama 700-8558, Japan; cometheart13@gmail.com (S.K.); 0203tkato@gmail.com (T.K.); teru_teru11_12@yahoo.co.jp (T.K.); nori.nishiwaki@gmail.com (N.N.); narusakat@gmail.com (T.N.); hiromaiaoi@gmail.com (H.S.); kyuki82@icloud.com (Y.K.); hkashima@s.okayama-u.ac.jp (H.K.); satorukc@okayama-u.ac.jp (S.K.); t_ohara@cc.okayama-u.ac.jp (T.O.);; 2Department of Pathology & Experimental Medicine, Okayama University Graduate School of Medicine, Dentistry, and Pharmaceutical Sciences, Okayama 700-8558, Japan; 3Center for Innovative Clinical Medicine, Okayama University Hospital, Okayama 700-8558, Japan

**Keywords:** cancer-associated fibroblast, chemotherapy, radiotherapy, esophageal cancer, tumor microenvironment

## Abstract

**Simple Summary:**

In esophageal cancer, the 5-year survival rate for patients who underwent esophagectomy with chemoradiotherapy is poor, with an estimated rate of 32.4%. One reason for this outcome is the low response rate to preoperative chemotherapy or radiotherapy. In this study, we revealed that low-dose chemotherapy or radiotherapy causes malignant phenotypic changes, such as cancer-associated fibroblasts, in normal fibroblasts. Furthermore, radiotherapy-resistant fibroblasts significantly increased the total number of tumors in the peritoneal cavity compared to the control group in peritoneal-disseminated tumor models. Thus, conventional cancer therapy has anti-therapeutic effects via activation of fibroblasts, and it is important to determine which modality of esophageal cancer treatment is selected or combined, recognizing that inappropriate radiotherapy and chemotherapy can lead to tumor resistance.

**Abstract:**

Esophageal cancer is one of the most aggressive tumors, and the outcome remains poor. One contributing factor is the presence of tumors that are less responsive or have increased malignancy when treated with conventional chemotherapy, radiotherapy, or a combination of these. Cancer-associated fibroblasts (CAFs) play an important role in the tumor microenvironment. Focusing on conventional cancer therapies, we investigated how CAFs acquire therapeutic resistance and how they affect tumor malignancy. In this study, low-dose chemotherapy or radiotherapy-induced normal fibroblasts showed enhanced activation of CAFs markers, fibroblast activation protein, and α-smooth muscle actin, indicating the acquisition of malignancy in fibroblasts. Furthermore, CAFs activated by radiotherapy induce phenotypic changes in cancer cells, increasing their proliferation, migration, and invasion abilities. In in vivo peritoneal dissemination models, the total number of tumor nodules in the abdominal cavity was significantly increased in the co-inoculation group of cancer cells and resistant fibroblasts compared to that in the co-inoculation group of cancer cells and normal fibroblasts. In conclusion, we demonstrated that conventional cancer therapy causes anti-therapeutic effects via the activation of fibroblasts, resulting in CAFs. It is important to select or combine modalities of esophageal cancer treatment, recognizing that inappropriate radiotherapy and chemotherapy can lead to resistance in CAF-rich tumors.

## 1. Introduction

Esophageal cancer is the 8th most common cancer worldwide, with over 604,000 patients, and is the 6th leading cause of cancer-related deaths. By 2020, more than 544,000 deaths were expected worldwide [1,2]. The two major subtypes of esophageal cancer are esophageal squamous cell carcinoma (ESCC) and esophageal adenocarcinoma (EAC). ESCC accounts for 90% of esophageal cancer cases worldwide and is predominant in East Asia, East Africa, and South America [2]. EAC is more common in developed countries than in developing countries. Currently, multidisciplinary treatment combining chemotherapy, radiotherapy, and surgery for advanced esophageal cancer is the standard therapy worldwide [3]. In addition, immunotherapies, such as immune checkpoint inhibitors have been approved by the Food and Drug Administration in the United States of America (FDA) for the treatment of esophageal cancer as a novel therapy. However, the 5-year survival rate of patients who undergo esophagectomy with chemoradiotherapy is estimated to be 32.4% [4]. One reason for this outcome is the low response rate to preoperative chemotherapy or radiotherapy (almost 40–60%) [4,5]. Therefore, improving the response rate to these conventional therapies is imperative for esophageal cancer therapy.

The tumor microenvironment (TME) influences therapeutic response and clinical outcomes. Numerous interactions between cancer cells and the TME have been reported to enable cancer cells to evade apoptosis by chemotherapy, resulting in chemoresistance mediated by soluble factors, cell adhesion, and immune response [6,7,8,9]. Cancer-associated fibroblasts (CAFs) compose a large population of stromal cells within the TME and provide important signals to assist tumor progression and escape antitumor therapy [10,11,12,13,14,15]. Interleukin (IL)-6 secreted by stromal cells, almost all CAFs, stimulates multiple myeloma cells to produce vascular endothelial growth factor, resulting in the activation of endothelial cells and fostering angiogenesis, leading to drug resistance [16,17]. In addition, CAFs induce cell adhesion-mediated drug resistance in tumors by alerting and remodeling the production of the extracellular matrix [18]. Moreover, increased hyaluronan production or matrix metalloproteinase 1 and 2 caused by CAFs promote drug resistance and metastasis in breast cancer and melanoma [18]. Thus, the acquisition of therapeutic resistance by CAFs is an important challenge for improving anticancer therapy.

CAFs can originate from a number of cell precursors and vary in tissues and organization. Originally, CAFs originated from localized resident fibroblast populations; however, they could also differentiate from mesenchymal stromal cells or stem cells. At the same time, the fibroblast lineage, epithelial cells, blood vessels, adipocytes, pericytes, and smooth muscle cells can differentiate into CAFs via endothelial cells [18,19]. Moreover, because tumors are heterogeneous biological systems and the origins of CAFs are diverse, the CAF phenotype, gene expression, and function are complex. Functional CAF subsets maintain unique profiles that create diversity in the TME; some CAF subsets do not affect tumor progression, whereas others modulate the TME in a pro-tumorigenic manner. Because of these characteristics, CAFs are considered a population in which the phenotype can be changed by various endogenous and exogenous stimuli [20].

In this study, we hypothesized that chemotherapy or radiotherapy could stimulate fibroblasts to become CAFs with malignant potential, resulting in the induction of therapeutic resistance in tumors. We evaluated whether conventional therapies, chemotherapy, or radiotherapy could activate fibroblasts and whether such activated fibroblasts contribute to tumor malignancy in vitro and in vivo. 

## 2. Materials and Methods

### 2.1. Cell Culture

Designated FEF3, a primary human esophageal fibroblast cell line, was isolated from the human fetal esophagus, as described previously [10]. NHLF, a normal human lung fibroblast cell line, was purchased from Cambrex Corporation (East Rutherford, NJ, USA), and WI-38, a human fetal lung fibroblast cell line, was purchased from the Health Science Research Resource Bank (Osaka, Japan). Human ESCCs (TE4 and TE8) and EACs (OE19 and OE33) were purchased from the Japanese Collection of Research Bioresources (JCRB) Cell Bank (Osaka, Japan). TE4-luc cells were transfected with luciferase using Lipofectamine^®^ 3000 (Invitrogen, Thermo Fisher Scientific, Waltham, MA, USA). Cancer cells were cultured in RPMI-1640 medium supplemented with 10% fetal bovine serum (FBS; Thermo Fisher Scientific) and 100 IU/mL penicillin and streptomycin. FEF3 and NHLF cells were cultured in DMEM (Sigma-Aldrich, St. Louis, MO, USA) with 10 % FBS and 100 IU/mL penicillin and streptomycin, WI-38 cells in MEM (Sigma-Aldrich) with 10 % FBS and 100 IU/mL penicillin and streptomycin at 37 °C in humidified air with 5% CO_2_ for no more than 30 passages.

### 2.2. Animals and Peritoneal Dissemination Model

Athymic female mice (BALB/c-nu/nu mice) that were approximately 6 weeks old were purchased from Clea (Tokyo, Japan). The animals were maintained under specific pathogen-free conditions in the animal laboratory at Okayama University. The protocols were approved by the Ethics Review Committee for Animal Experimentation of Okayama University, Okayama, Japan (approval no. OKU-2018790).

### 2.3. Reagents

Cisplatin (CDDP, Randa) was purchased from Nippon Kayaku (Tokyo, Japan), and 5-fluorouracil (5-FU) was purchased from KYOWA KIRIN (Tokyo, Japan). The reagents were dissolved in phosphate-buffered saline (PBS).

### 2.4. Cell Viability Assay

Cell viability was evaluated using the Cell Proliferation Kit II (XTT; Roche Molecular Biochemicals, Mannheim, Germany), as described previously [13]. Fibroblasts or cancer cells were treated with the indicated concentrations of anticancer drugs (5-FU and CDDP). Approximately 72 h after chemotherapy, cell proliferation was measured according to the manufacturer’s protocol. For radiotherapy, sensitivity in fibroblasts or cancer cells was estimated using the XTT assay five days after several doses (0, 5, 10, 25, and 50 Gy) of single-fraction radiation. For quantitative evaluation, cancer cells were seeded in multi-well plates and incubated for 24 h in RPMI-1640 medium. The medium was replaced with the conditioned medium of irradiated fibroblast (Day 0). The cancer cells were cultured in conditioned medium, and cellular proliferation was evaluated by the XTT assay every 2 days from day 0 to day 10.

### 2.5. Fibroblast Cell Proliferation under Fractionated Irradiation

Fibroblasts were seeded in a multi-well plate (Day 0) and treated with several doses (0, 4, 8, 20 Gy) of irradiation in 4 fractions on days 1, 2, 4, and 5. Fibroblasts were dissociated from the plate with trypsin after washing by PBS and cell numbers were counted with a hemocytometer on days 2, 4, 6, 8, and 10.

### 2.6. Treatment by Chemotherapy and Radiotherapy

For chemotherapy, fibroblasts were seeded 24 h before treatment. Then, these cells were treated with approximately IC50 of each anticancer drug (7.5 µM for CDDP; 50 µM for 5-FU) or PBS for 72 h. For radiotherapy, fibroblasts were seeded on day 0 and treated with 16 Gy ionized radiation in 8 fractions of 2 Gy irradiation from day 1 to day 12, and 48 h later, the treated cells were used. During treatment, fibroblasts were passaged at 80% confluence in a culture flask.

### 2.7. Immunofluorescence Microscopy for α-SMA and FAP in Fibroblasts

Fibroblasts were cultured on chambered coverslips for 72 h. For staining with alpha smooth muscle actin (α-SMA), the treated fibroblasts were fixed and permeabilized with methanol for 30 min at 4 °C. Cells were incubated with a 1:300 dilution of primary antibody against α-SMA (ab5694, Abcam, Cambridge, UK) at 4 °C overnight, and then incubated with a 1:200 dilution of secondary antibody Alexa568 (Thermo Fisher Scientific) at room temperature for 2 h. For staining with fibroblast activation protein (FAP), the treated fibroblasts were fixed with 4% paraformaldehyde for 15 min. Cells were incubated with a 1:300 dilution of primary antibody against FAP (MAB3715, R&D Systems, Minneapolis, MN, USA) at 4 °C overnight, and then incubated with a 1:200 dilution of secondary antibody Alexa488 (Thermo Fisher Scientific) at room temperature for 2 h. Cells were incubated with 0.1μg/mL DAPI, and the slides were imaged with a confocal laser scanning microscope (LMS780, ZEISS, Oberkochen, Germany).

### 2.8. Flow Cytometry for α-SMA and FAP in Fibroblasts

Fibroblasts were washed with PBS and dissociated from the vessels with trypsin. The cells were fixed with 4% paraformaldehyde at 37 °C for 10 min. The cells were incubated with 1:100 dilution of primary antibody against FAP (MAB3715, R&D Systems) or IgG1 (sc-3877, Santa Cruz Biotechnology, Dallas, TX, USA) for 30 min at room temperature and incubated with 1:200 dilution of secondary antibody Alexa647 (A21237, Thermo Fisher Scientific) for 30 min at room temperature in the dark. The cells were permeabilized with a permeabilization buffer (#421002, BioLegend, San Diego, CA, USA) and a 1:100 dilution of antibody against α-SMA conjugated with FITC (ab8211, Abcam, UK) or IgG2a conjugated with FITC (#400210, BioLegend) for 30 min at room temperature in the dark. The cells were analyzed for two antigens simultaneously using a BD FACSLyric flow cytometer (BD Biosciences, San Jose, CA, USA), and the obtained data were estimated using BD FACSuite software (BD Biosciences). The analyzed cells were confirmed to exclude dead cells using Zombie Aqua (#423101, BioLegend).

### 2.9. Conditioned Medium of Resistant Fibroblast Induced by Radiotherapy

Fibroblasts treated with (or without) 16 Gy ionized radiation in 8 fractions of 2 Gy irradiation for 12 days were passaged in DMEM with 2% FBS and cultured for 48 h. The supernatant harvested from the centrifuged medium was used as the conditioned medium for irradiated (or normal) fibroblasts. 

### 2.10. Migration and Invasion Assay

The effects of the conditioned medium of irradiated fibroblasts on cancer cell migration and invasion were estimated by the Boyden chamber technique using a BD BIoCoat Matrigel invasion chamber (8 µm pore size) or Migration Chamber (8 µm pore size) according to the manufacturer’s protocol. A total of 1.0 × 10^5^ cells per well for TE4, TE8, and OE33 cells and 2.0 × 10^5^ cells for OE19 cells were seeded and analyzed.

### 2.11. ELISA

The concentration of IL-6 in the conditioned medium of irradiated (or normal) fibroblasts was assessed using the Quantikine ELISA human IL-6 Immunoassay (R&D Systems) according to the manufacturer’s protocol.

### 2.12. Bioluminescence Images

BALB/c-nu/nu mice were administered 0.5 × 10^6^ TE4-luc cells and 2.0 × 10^6^ FEF3 cells with or without 16 Gy irradiation in 8 fractions. To evaluate the growth of peritoneal metastasis over time, mice were administered D-luciferin (VivoGlo™ Luciferin, In Vivo Grade, Promega Corporation, Madison, WI, USA) and imaged under isoflurane anesthesia every 7 days. Bioluminescence images were obtained using an IVIS Lumina imaging system (Xenogen IVIS Lumina II; Caliper Life Sciences, Hopkinton, MA, USA), and image analysis and bioluminescence quantification were performed using Living Image software.

### 2.13. Statistical Analysis

Statistical analysis was performed using JMP software (SAS Institute, Cary, NC, USA). For two-group comparisons, the Mann–Whitney test or unpaired *t*-test was used. Statistical significance was set at *p* < 0.05.

## 3. Results

### 3.1. Cancer Cells and Fibroblasts Damaged by Chemotherapy and Radiotherapy in a Dose-Dependent Manner

To evaluate the efficacy of chemotherapies using 5-FU or CDDP for cancer cells and fibroblasts, cell viability was tested using the XTT assay 72 h after treatment in vitro. Human-derived fibroblasts (FEF3, NHLF, and WI-38) were significantly damaged by both agents in a dose-dependent manner (Figure 1A). Esophageal cancer cell lines (TE4, TE8, and OE33) were destroyed by both drugs (Figure 1B). From these results, the IC50 concentrations against fibroblasts were calculated as follows: CDDP, 7.5 µM and 5-FU, 50 µM. Next, we assessed the efficacy of radiotherapy in fibroblasts and cancer cells. In TE4 cells, the efficacy of radiotherapy increased in an X-ray-dose dependent manner; however, cells damaged by radiotherapy were minimally detectable in FEF3 cells (Figure 1C). Furthermore, FEF3 cell proliferation was inhibited at a total X-ray dose of 4 Gy or more by counting cell proliferation (Figure 1D). Similar results were also obtained using WI-38 fibroblasts treated with radiotherapy (Appendix A). These results suggest that radiotherapy did not destroy fibroblasts, but the mitogenic activity was suppressed.

### 3.2. Low-Dose Anticancer Agents Induce the Malignant Activation of Fibroblasts

Based on previous results, chemotherapy at the IC50 was defined as a low-dose therapeutic concentration in this study. Immunofluorescence images were acquired to investigate the phenotypic changes in normal fibroblasts induced by chemotherapy. The treatment regimen is shown in Figure 2A. Treated fibroblasts were collected 72 h after exposure to anticancer agents. In FEF3 cells treated with both 5-FU and CDDP, the fluorescence intensity of FAP, which is a surface marker of CAFs, was increased compared to that in untreated fibroblasts (Figure 2B). Furthermore, the intracellular expression of α-SMA was elevated. Similar results were observed for treated NHLF and WI-38 cells. Quantitative analyses by flow cytometry showed a 2–4-fold increase in FAP and α-SMA expression in treated fibroblasts compared to untreated fibroblasts in all cell lines (Figure 2C,D). Additionally, Western blot analysis revealed an increase in FAP expression in treated NHLF cells (Appendix A). These results suggest that inadequate chemotherapy for fibroblasts leads to the acquisition of malignant potential in CAFs.

### 3.3. Radiotherapy Intensively Caused Fibroblasts to Transform into CAFs 

We also evaluated phenotypic changes in fibroblasts using X-rays for cancer treatment. First, we verified which dose of radiation would activate normal fibroblasts the most (Appendix A). Radiation-dose-dependent fibroblast activation was observed up to a dose of 16 Gy; however, no additive effect was observed above this dose. Based on these results, in this study, 16 Gy in total (2 Gy each for 8 days) was irradiated with fibroblasts as low-dose radiotherapy, and one line dose was set at 2 Gy according to standard radiotherapy in clinical practice. The treatment regimen is shown in Figure 3A. Treated fibroblasts highly expressed FAP on the cell surface and α-SMA in the intracellular space compared to untreated fibroblasts in all three cell lines (Figure 3B). Furthermore, in both FAP and α-SMA expression, the fluorescence signals after radiotherapy were more intense than those after treatment with chemotherapy, which were 4–10-fold higher than those of the controls in all human-derived fibroblasts via flow cytometric analysis (Figure 3C,D). This result indicates that radiotherapy strongly induces CAF activation of normal fibroblasts compared to chemotherapy. Thus, fibroblasts treated with X-ray irradiation were more activated than those treated with chemotherapy and had malignant potential.

### 3.4. X-ray-Irradiated Fibroblasts Promoted Cancer Cell Proliferation, Invasion, and Migration In Vitro

To explore the impact of radiation-treated FEF3 fibroblasts (RFs) on esophageal cancer cells (TE4, TE8, OE19, and OE33), the proliferation of cancer cells was tested. The treatment regimen is shown in Appendix A. The three esophageal cancer cell lines (TE4, OE19, and OE33) cultured in conditioned media made from RFs (CM-RF) showed significantly enhanced proliferation compared to those cultured in conditioned media made from FEF3 normal fibroblasts (CM-NF, control; Figure 4A). In contrast, TE8 cells were unaffected by CM-RF. The acquisition of radiotherapeutic resistance in cancer cells by RFs was evaluated using an XTT assay (Appendix A). Cancer cells cultured in CM-RF (TE4, OE19, and OE33) showed higher viability after irradiation with X-rays than those cultured in CM-NF, but no significant difference was observed. Additionally, we performed migration and invasion assays to evaluate the effect of CM-RF on other malignant potentials (Figure 4B). The migration assay showed that the number of migrated cells in the CM-RF groups increased by 2–4-fold compared to cancer cells in the CM-NF groups (Figure 4C). In addition, the number of invading cancer cells treated with CM-RF significantly increased in TE4 and OE33 esophageal cancer cell lines (Figure 4D,E). These results demonstrate that RF-CM has a malignant phenotype in cancer cells. Furthermore, the concentration of IL-6, an inflammatory cytokine, in each CM was significantly higher in CM-RF than in CM-NF, suggesting acquisition of cancer cell malignancy through secretion of IL-6 in CM-RF. (Appendix A). 

### 3.5. Fibroblasts Irradiated with X-rays Promoted Peritoneal Dissemination in Esophageal Tumor Models

To evaluate the effect of RFs on the migration or metastasis of cancer cells in vivo, luciferase-transfected TE4 (TE4-luc) cells were co-inoculated with FEF3 cells treated with or without radiation into the abdominal cavity of athymic mice. The treatment regimen is shown in Figure 5A. We obtained fluorescence images and calculated the total luciferase fluorescence per week. On the 28th day after inoculation, luciferase activity in the TE4+RF group was significantly higher than that in the TE4+NF group (Figure 5B,C). No change in body weight was detected between the two groups during the entire observation period (Figure 5D). To evaluate the total tumor weight and number of metastases, the mice were sacrificed on the 29th day, and visible peritoneal disseminations were harvested (Figure 5E). The total number of tumor nodules in the abdominal cavity was significantly higher in the TE4+RF group than in the TE4+NF group, and the same tendency was observed for the total tumor weight of the peritoneal metastatic nodules (Figure 5F). These results demonstrated that activated fibroblasts irradiated with X-rays enhanced peritoneal dissemination in an in vivo model.

## 4. Discussion

We demonstrated that low-dose chemotherapy or radiotherapy stimulated fibroblasts, resulting in the induction of CAFs overexpressing FAP or α-SMA. This activation was observed more strongly in radiotherapy, and these stimulated fibroblasts caused greater tumor progression, migration, and invasion than the untreated fibroblasts. Furthermore, lL-6 production from X-ray-irradiated fibroblasts was significantly increased, suggesting activation of CAFs and tumor progression [11,17]. Additionally, peritoneal dissemination was enhanced by intraperitoneal co-inoculation of cancer cells and fibroblasts treated with low-dose radiotherapy. Thus, these results suggest that inadequate conventional cancer therapies induced CAFs, and malignancy in cancer cells was also increased by activated CAFs.

We previously reported that CAFs play essential roles in the TME and cause malignant tumor progression [10,11,12,13,17]. CAFs have also been shown to acquire resistance to chemotherapy or radiotherapy [14]. In this study, we revealed that drugs or cancer therapies originally designed to treat malignant tumors had the opposite effect of promoting the malignant potential of the TME, especially CAFs. Increased CAF markers indicate the acquisition of a malignant phenotype, and functional malignant enhancement of CAF was observed, given the elevation of IL-6, which is known to be produced from CAFs, regulated immunosuppressive tumor-infiltrating lymphocyte population in CAF-rich tumors [11,17]. This event is unlikely to occur with adequate doses of chemotherapy or radiotherapy. However, considering the pharmacokinetics and pharmacodynamics of the drug in vivo, the predicted drug concentration in the tumor may not be insufficient, suggesting that inadequate treatment could induce phenotypic changes in CAFs. These findings may be a step toward solving the clinical problems associated with chemotherapy or radiotherapy resistance.

In this study, fibroblasts were more activated as CAFs by radiotherapy than by chemotherapy. One reason could be the difference in the response to each cancer therapy. Normal fibroblasts were damaged and destroyed by chemotherapy in a dose-dependent manner, whereas radiotherapy induced cell growth inhibition but not cell death. This led to more resistant fibroblasts being derived from radiotherapy than from chemotherapy. Another reason is that CAFs are reversible, and loss of stimulation after chemotherapy may lead to fibroblast inactivation. In contrast, radiotherapy was found to maintain the increased expression of CAF markers and malignant potential, as the changes after X-ray irradiation were sustained. Therefore, in a clinical setting, more attention should be paid to the acquisition of radiotherapeutic resistance in CAFs.

Additionally, the effect on CAFs depends on the type of anticancer drug used. For instance, docetaxel had a weaker effect on fibroblasts than 5-FU or CDDP. This finding is concordant with previous reports that triplet chemotherapy with docetaxel, 5-FU, and CDDP is more effective than doublet chemotherapy with 5-FU and CDDP [21,22]. In the future, drug combinations or treatment strategies may be determined based on the status.

To solve this problem, which involves unintended fibroblast activation using conventional therapies, it is necessary to eliminate treatment-resistant CAFs in tumors. FAP-targeted near-infrared photoimmunotherapy (NIR-PIT) has been developed [13,14,15] and has the advantage that the therapeutic effect is observed in the area irradiated by NIR light and does not affect the non-irradiated site [23]. After treatment with conventional therapies, CAFs strongly expressed FAP, suggesting that FAP-targeted NIR-PIT has the potential to selectively delete local CAFs in the tumor. Thus, adding treatment to the TME, especially CAFs, to conventional therapy could theoretically be effective.

This study has several limitations. First, we verified that only one condition, chemotherapy and radiotherapy, was validated for the acquisition of therapeutic resistance in CAFs. In this study, the IC50 concentration was applied for chemotherapy and 16 Gy for radiotherapy, in which normal fibroblasts were the most activated. The definition of insufficient therapeutic status can vary depending on the tumor grade and host condition, and no exact definition exists. Therefore, different conditions may lead to different levels of therapeutic resistance. Second, human-derived cancer cells and fibroblasts were used to simulate clinical settings. We successfully captured time-course images using bioluminescence imaging; however, the influence of tumor immunity could not be evaluated in this model. The acquisition of conventional therapeutic resistance may affect tumor immunity. Additionally, the level of IL-6 was measured to evaluate the potential tumor malignancy in the conditioned media treated with radiotherapy because we previously reported that IL-6 is the most significantly altered humoral factor when normal fibroblasts transform into CAFs (Appendix A) [11,17]. However, CAF-derived conditioned media contains a variety of substances; therefore, more investigation will be necessary to elucidate the exact mechanism underlying these associations.

## 5. Conclusions

Inadequate chemotherapy and radiotherapy induced phenotypic changes in fibroblasts, resulting in CAFs and increased tumor malignancy. In a clinical setting, it is important to select the appropriate dose of these therapies to improve the response to conventional therapy. Furthermore, the combined use of conventional and CAF-targeted therapies is expected to provide synergistic effects.

## Figures and Tables

**Figure 1 cancers-15-02971-f001:**
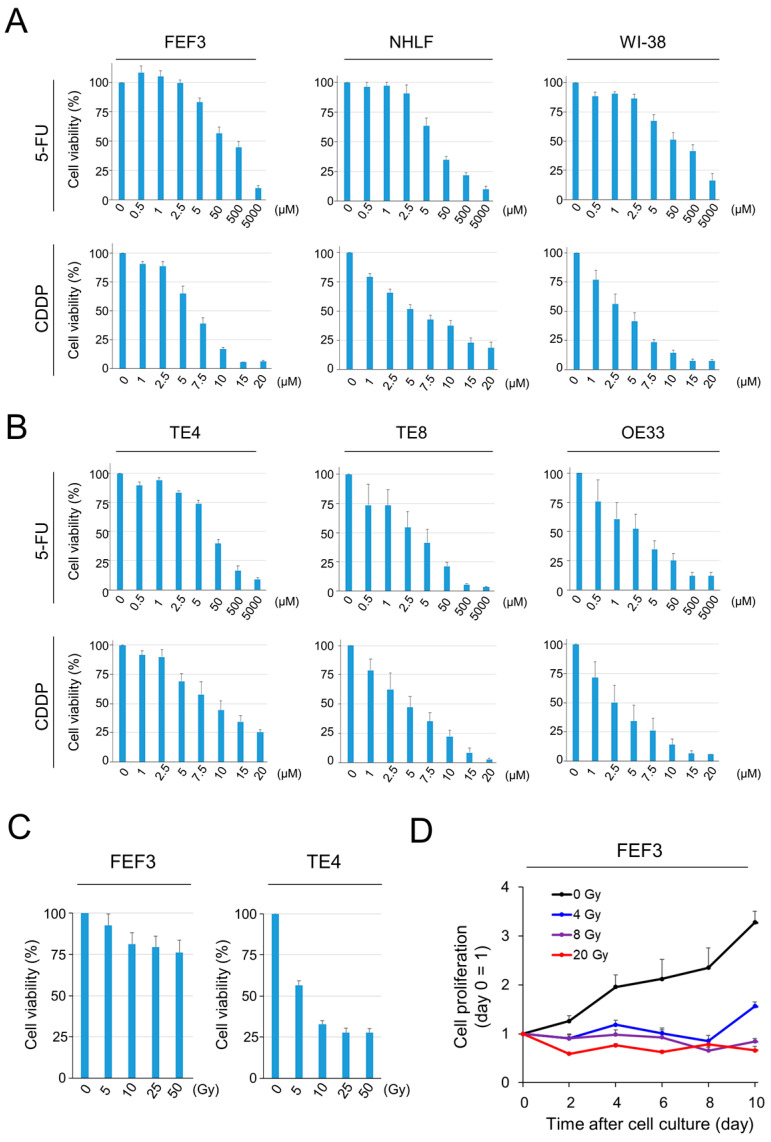
Efficacy of chemotherapy or radiotherapy against cancer cells and fibroblasts. (**A**,**B**) Viability of human-derived fibroblasts (**A**) or cancer cells (**B**) induced by chemotherapy (CDDP or 5-FU) was measured using XTT metabolic activity. The control group (100 %) did not receive chemotherapy (*n* = 4. Mean ± SD). (**C**) Viability of TE4 and FEF3 cells induced by radiotherapy measured using XTT (*n* = 4. Mean ± SD). (**D**) Relative proliferation curves of FEF3 cells treated by radiotherapy. Day 0 is set as the control (*n* = 4; mean ± SD). In (**A**–**C**), the representative examples from five experiments were shown, and the representative example from three experiments was shown in (**D**).

**Figure 2 cancers-15-02971-f002:**
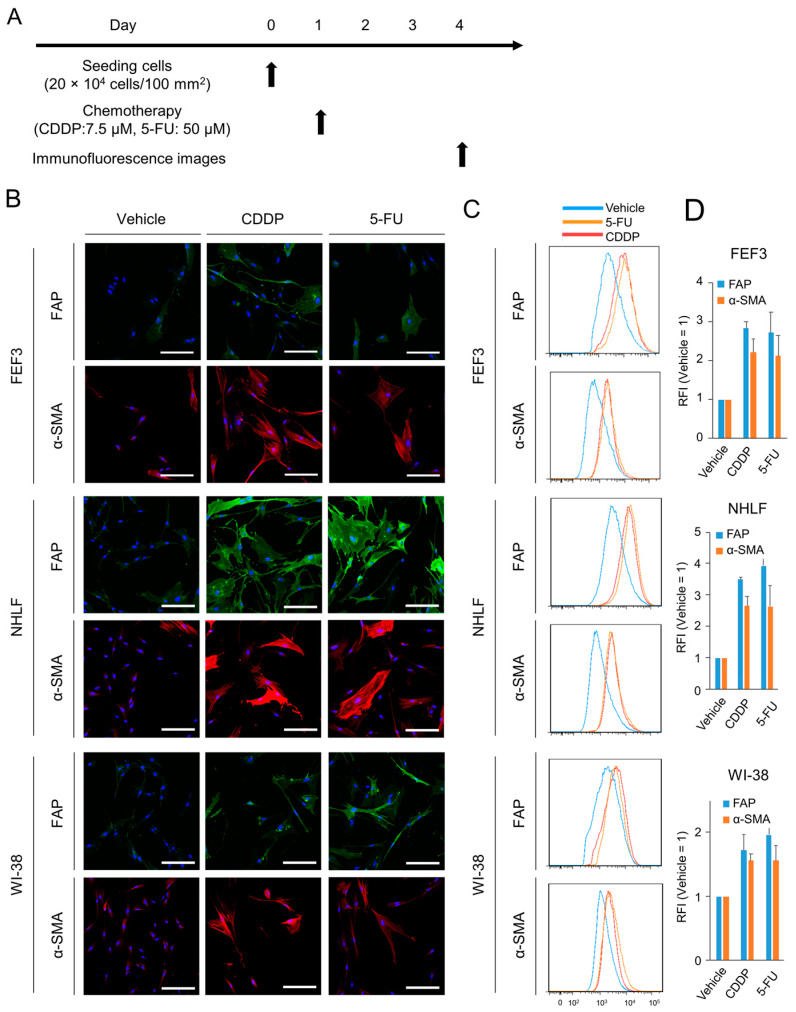
Phenotypic changes in normal fibroblasts after chemotherapy. (**A**) Treatment schedule. (**B**) Fluorescent microscopic images of human fibroblasts (FEF3, NHLF, and WI-38) after chemotherapy (CDDP or 5-FU). FAP (green), α-SMA (red), and DAPI (blue) staining is shown. Scale bar, 20 µm. The representative examples from four experiments were shown. (**C**,**D**) Flow cytometric analysis of FAP and α-SMA expression in the three fibroblast cell lines. Representative flow cytometric histograms are shown (**C**), and the relative fluorescence intensity (RFI) was calculated ((**D**), *n* = 4. Mean ± SD).

**Figure 3 cancers-15-02971-f003:**
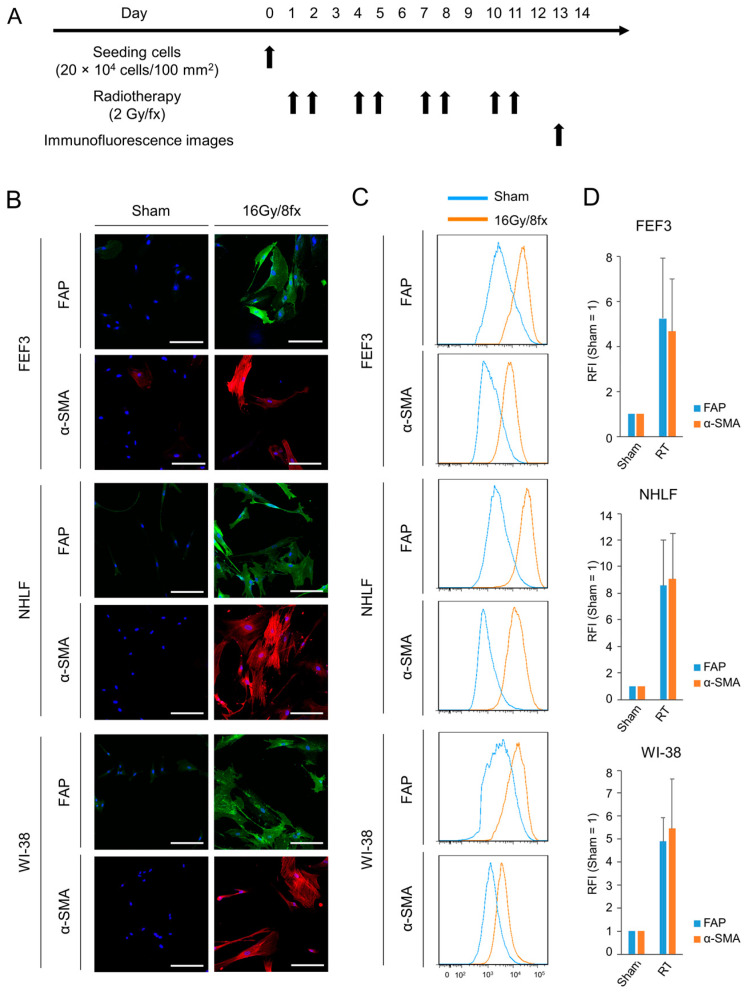
Phenotypic changes in fibroblasts following radiotherapy. (**A**) Treatment schedule. (**B**) Fluorescent microscopic images of human-derived fibroblasts (FEF3, NHLF, and WI-38) after radiotherapy (16 Gy/8 fx). FAP (green), α-SMA (red), and DAPI (blue) staining is shown. The Sham group is used as the control group. Scale bar, 20 µm. The representative examples from four experiments were shown. (**C**,**D**) Flow cytometric analysis of FAP and α-SMA expression in the three fibroblast cell lines. Representative flow cytometric histograms are shown (**C**), and the RFI was calculated ((**D**), *n* = 4. Mean ± SD).

**Figure 4 cancers-15-02971-f004:**
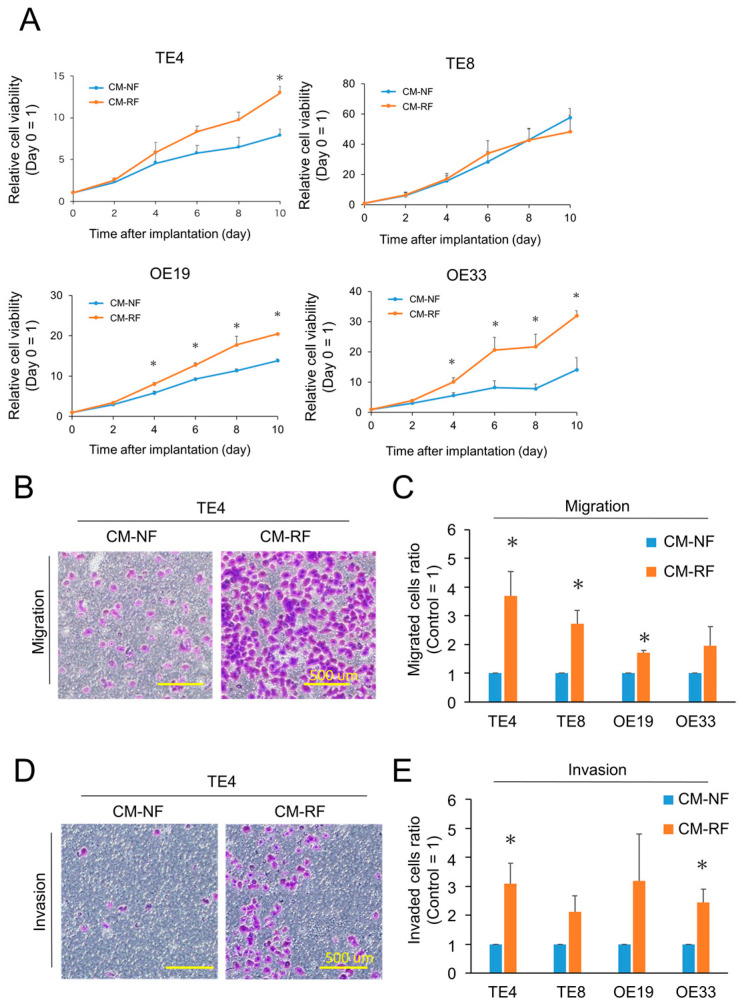
Cancer cells stimulated with resistant fibroblasts exhibited malignant characteristics. (**A**) Relative proliferation curves of cancer cells treated with conditioned media made of normal fibroblasts (CM-NF) or resistant fibroblasts (CM-RF). Day 0 was set as the control (*n* = 4, mean ± SD, unpaired *t*-test; * *p* < 0.05). (**B**,**C**) Migration assay of esophageal cancer cells cultured with CM-NF or CM-RF. (**B**) Representative migrated TE4 esophageal cancer cells are shown. Scale bar = 500 µm. The representative example from four experiments was shown. (**C**) Relative migrated cells (control = CM-NF) were calculated and shown for four esophageal cancer cells. (*n* = 4, mean ± SD, unpaired *t*-test; * *p* < 0.05). (**D**,**E**) Invasion assay of esophageal cancer cells cultured with CM-NF or CM-RF. (D) Representative invaded TE4 esophageal cancer cells are shown. Scale bar = 500 µm. The representative example from four experiments was shown. (**E**) Relative invaded cells (control = CM-NF) were calculated and are shown for four esophageal cancer cells. (*n* = 4, mean ± SD, unpaired *t*-test; * *p* < 0.05).

**Figure 5 cancers-15-02971-f005:**
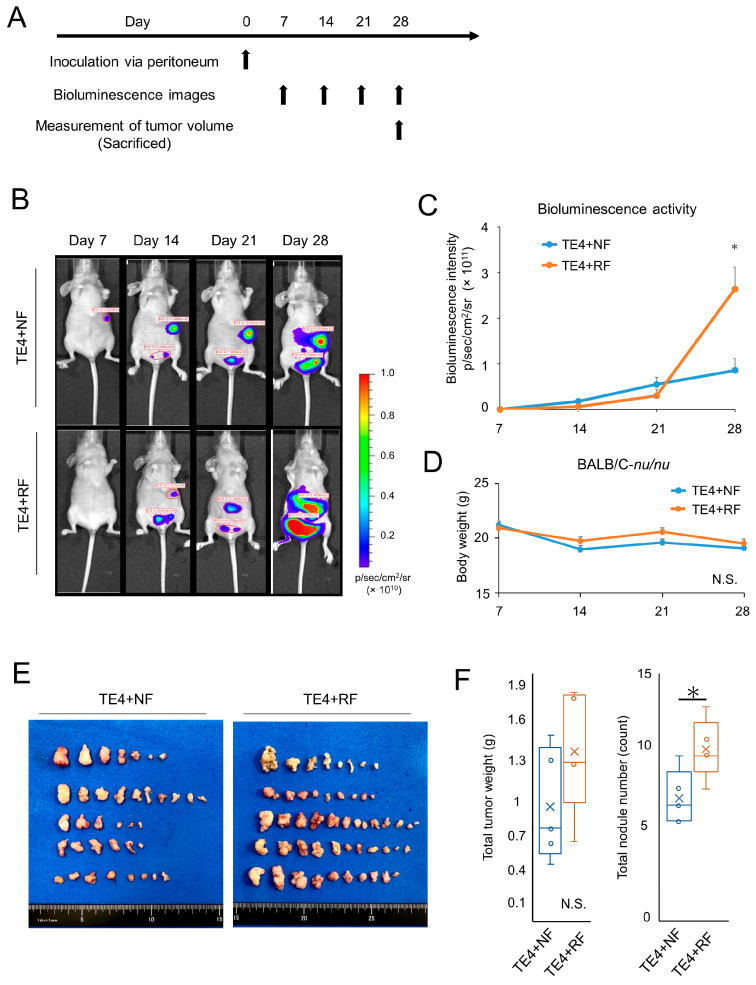
Resistant fibroblasts treated by radiotherapy enhanced tumor dissemination in vivo models. (**A**) Treatment regimen. (**B**) Bioluminescence images of TE4-luc dissemination mouse models. (**C**) Luciferase activity calculated from the bioluminescence intensity (*n* = 4; mean ± SD, unpaired *t*-test; *, *p* < 0.05). (**D**) Mean body weight of each group (*n* = 4; mean ± SD; N.S., not significant). (**E**) Representative harvested disseminated tumors per mouse in each group. (**F**) Total weights or counts of harvested disseminated tumors were calculated for each group, and box and whisker plots are shown (*n* = 4; unpaired *t*-test; * *p* < 0.05, N.S., not significant).

## Data Availability

The data generated in this study are available within the article and its Appendix A files.

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
