# Peer review of "Conventional Cancer Therapies Can Accelerate Malignant Potential of Cancer Cells by Activating Cancer-Associated Fibroblasts in Esophageal Cancer Models"

_cancers, 2023, doi:10.3390/cancers15112971_

Round 1
Reviewer 1 Report
Komoto and collaborators investigated the effects of insufficient chemotherapy or radiotherapy on normal fibroblasts, leading to their activation to pro-tumoral CAFs.
The experimental data are original and well organized; the reading is fluent. Significant consequences may arise from these studies impacting the clinical setting.
Minor issue:
The inflammatory cytokine IL-6 was proposed to contribute to cancer cell malignancy. What about other possibilities? The conditioned media can contain a variety of molecules or vesicles. The author can better address this point.
Author Response
Komoto and collaborators investigated the effects of insufficient chemotherapy or radiotherapy on normal fibroblasts, leading to their activation to pro-tumoral CAFs. The experimental data are original and well organized; the reading is fluent. Significant consequences may arise from these studies impacting the clinical setting. Minor issue: The inflammatory cytokine IL-6 was proposed to contribute to cancer cell malignancy. What about other possibilities? The conditioned media can contain a variety of molecules or vesicles. The author can better address this point. →Thank you for your comments. We completely agree with the potential interaction of cancer cells and various cytokines, chemokines, and small particle materials. In our previous report (T. Kato et al., Clin Can Res. 2018), we demonstrated that IL-6 was the most significantly altered humoral factor when normal fibroblasts transformed into CAFs. Therefore, we initially investigated IL-6, but further investigation is necessary to elucidate the exact mechanism underlying these associations. We have modified and added the following sentence to the discussion section of limitation paragraph; Additionally, the level of IL-6 was measured to evaluate the potential tumor malignancy in the conditioned media treated with radiotherapy because we previously reported that IL-6 is the most significantly altered humoral factor when normal fibroblasts transform into CAFs (Figure S6) [11,17]. However, CAF-derived conditioned media contains a variety of substances; therefore, more investigation will be necessary to elucidate the exact mechanism underlying these associations
Reviewer 2 Report
The study by Komoto et al. investigates the influence of chemo- and/or radiotherapy on esophageal cancer cells and fibroblasts. The applied treatment regimen on fibroblasts resulted in upregulation of markers for cancer-associated fibroblasts in vitro. Conditioned media of treated fibroblasts enhanced cancer phenotypes of esophageal cancer cell lines in vitro and increased peritoneal spread in vivo.
The study is interesting. The following points should be considered.
Major points
1) The terminology “insufficient therapy” is problematic as it can easily be misunderstood as a treatment not according to guidelines. Since it is not straight forward to transfer concentrations of chemotherapy in vitro to what is applied to patients and the same is true for radiotherapy, the authors should discuss carefully how the in vitro results transfer to patients.
2) For all experiments, the number of replicates are mentioned. However, it has to be also mentioned, how many experiments have been performed, e.g. one representative out of three experiments.
3) Can the authors exclude that the fibroblasts increased proliferation upon radiation? I admit, this would not be expected but it would bias a number of experiments, i.e. the ELISA of IL-6 would have to be corrected for cell numbers and similarly correction would be required for the experiments with conditioned media.
4) There is a discrepancy between the flow cytometry plots in Fig. 3C and the bar charts in Fig. 3D for NHLF and WI-38. Is the red/blue color coding for the flow sort plots wrong?
5) The experiments in Fig. 1 C and D should be done for all 6 cell lines.
6) The statement in lines 279 to 281 that cancer cells cultured with CM-RF showed higher viability seems to be based on Supp Fig. S4. I do not see statistically significant differences that justify that statement.
7) Which fibroblast cell line(s) were used for experiments in Fig. 4? FEF3? The experiments in Fig. 4 should be performed with at least two different fibroblast cell lines to support the findings.
8) The institutional Review Board Statement did contain only the default text by Cancers and remains to be written for this project.
Minor points
1) What were the numbers of cells that were seeded for the migration and invasion assays?
2) The sentence in lines 256 to 260 should be rephrased as it currently sounds like radiation has a 4-10 fold stronger effect compared to chemotherapy.
3) Lines 289 to 290 “… suggesting that IL-6 contributed to cancer cell malignancy” is not correct. It is rather that CM-RF might contribute to cancer cell malignancy through induction of IL-6 expression.
4) Figure 5: It seems that four mice have been used. The arrangement of the tumors in Fig. 5E suggests that maybe 5 mice were used? Please clarify.
5) Supp Fig. S6 was not included in my documents.
Minor editing will improve the manuscript.
Author Response
Comments and Suggestions for Authors: Reviewer 2
The study by Komoto et al. investigates the influence of chemo- and/or radiotherapy on esophageal cancer cells and fibroblasts. The applied treatment regimen on fibroblasts resulted in upregulation of markers for cancer-associated fibroblasts in vitro. Conditioned media of treated fibroblasts enhanced cancer phenotypes of esophageal cancer cell lines in vitro and increased peritoneal spread in vivo.
The study is interesting. The following points should be considered.
Major points
1) The terminology “insufficient therapy” is problematic as it can easily be misunderstood as a treatment not according to guidelines. Since it is not straight forward to transfer concentrations of chemotherapy in vitro to what is applied to patients and the same is true for radiotherapy, the authors should discuss carefully how the in vitro results transfer to patients.
→Thank you for the appropriate suggestion. As the main point of this paper, we demonstrated that “low-dose” conventional therapies, which do not destroy target cells, could adversely affect normal cells. As the reviewer mentioned, we agree that the terminology 'insufficient therapy' is inappropriate. We have changed the word “insufficient” to “low-dose” in the Results and Discussion sections.
2) For all experiments, the number of replicates are mentioned. However, it has to be also mentioned, how many experiments have been performed, e.g. one representative out of three experiments.
→As the reviewer mentioned, it is important to indicate the number of replications. We added the replicating information in Figure Legends each as follows;
Figure 1 (A-D). In figure 1A-C, the representative examples from five experiments were shown, and the representative example from three experiments was shown in figure 1D.
Figure 2B. The representative examples from four experiments were shown.
Figure 3B. The representative examples from four experiments were shown.
Figure 4B, D. the representative example from four experiments was shown.
3) Can the authors exclude that the fibroblasts increased proliferation upon radiation? I admit, this would not be expected but it would bias a number of experiments, i.e. the ELISA of IL-6 would have to be corrected for cell numbers and similarly correction would be required for the experiments with conditioned media.
 →Thank you for pointing that out. To calculate the relative IL-6 concentrations, the values of IL-6 concentration per 1.0 x 104 cells were obtained to exclude inequality. To avoid mistakes, the following sentence was added in figure legend in Figure S5;
The level of IL-6 in each supernatant of conditioned media was measured using Quantikine ELISA human IL6 Immunoassay (R & D systems). Each data was acquired as IL-6 levels per, then relative IL-6 concentration was calculated.
4) There is a discrepancy between the flow cytometry plots in Fig. 3C and the bar charts in Fig. 3D for NHLF and WI-38. Is the red/blue color coding for the flow sort plots wrong?
→Thank you for pointing out his serious error. We corrected these figures in Figure 3.
5) The experiments in Fig. 1 C and D should be done for all 6 cell lines.
 →As indicated in the title, the aim of this study was to clarify the association between CAFs and conventional cancer therapies in esophageal cancer models. To address this issue, we used TE4 and FEF3 cells. TE4 cells were obtained from human esophageal cancer patients, while FEF3 cells were derived from human esophagus. Experiments using these cells derived from the human esophagus are well-established in our institution and have already been reported in several papers (Watanabe et al., Cancer Biology & Therapy 2019; Katsube et al., Sci Rep 2021; Kashima et al., International Journal of Cancer 2019). Therefore, all experiments were validated using these cells. To support our findings, we also demonstrated that similar responses occur in other esophageal cancer cells (TE8 or OE19) and lung-derived fibroblasts (NHLF or WI38) in Figures 1-4. In fact, similar results were obtained for the efficacy of radiotherapy using WI-38 fibroblasts. These results were added the sentence in the result section and supplementary figure S1 as follows;
Similar results were also obtained using WI-38 fibroblasts treated with radiotherapy (Figure S1).
Figure S1. Efficacy of radiotherapy against WI-38 fibroblasts.
(A) Viability of WI-38 cells induced by radiotherapy measured using XTT (n = 4. Mean ± SD). The representative example from three experiments was shown. (B) Relative proliferation curves of WI-38 cells treated by radiotherapy. Day 0 is set as the control (n = 4; mean ± SD). The representative example from three experiments was shown.
6) The statement in lines 279 to 281 that cancer cells cultured with CM-RF showed higher viability seems to be based on Supp Fig. S4. I do not see statistically significant differences that justify that statement.
→As the reviewer pointed it out, there are no significant difference in these experiments. To avoid misleading readers, we modified the sentence as follows;
Cancer cells cultured in CM-RF (TE4, OE19, and OE33) showed higher viability after irradiation with X-rays than those cultured in CM-NF, but no significant difference was observed.
7) Which fibroblast cell line(s) were used for experiments in Fig. 4? FEF3? The experiments in Fig. 4 should be performed with at least two different fibroblast cell lines to support the findings.
→Thank you for identifying the missing explanation. In Figure 4, we used CM-NF and CM-RF, which were derived from FEF3 cells. We have added this information in the Results section (3.4.) and the figure legend in Figure 4. As described previously, we aimed to simulate the tumor microenvironment of esophageal cancer to elucidate the correlation between CAFs and conventional therapies. Therefore, we used TE4 and FEF3 cells in these experiments.
8) The institutional Review Board Statement did contain only the default text by Cancers and remains to be written for this project.
 → We apologize for this mistake. Now, we corrected our declaration in this section as follows;
The animal study protocol was approved by the Ethics Review Committee for Animal Experimentation of Okayama University, Okayama, Japan (approval no. OKU-2018790).
Minor points
1) What were the numbers of cells that were seeded for the migration and invasion assays?
 → 1.0 × 105 cells per well for TE4, TE8, and OE33 cells, and 2.0 × 105 cells for OE19 cells were seeded and analyzed. We added this information in Material and Methods section.
2) The sentence in lines 256 to 260 should be rephrased as it currently sounds like radiation has a 4-10 fold stronger effect compared to chemotherapy.
→We added the sentence as follows;
This result indicates that radiotherapy strongly induces CAF activation of normal fibroblasts compared to chemotherapy.
3) Lines 289 to 290 “… suggesting that IL-6 contributed to cancer cell malignancy” is not correct. It is rather that CM-RF might contribute to cancer cell malignancy through induction of IL-6 expression.
→ Thank you for correcting our mistake. We corrected the sentence as follows;
,suggesting acquisition of cancer cell malignancy through secretion of IL-6 in CM-RF.
4) Figure 5: It seems that four mice have been used. The arrangement of the tumors in Fig. 5E suggests that maybe 5 mice were used? Please clarify.
→These figures show representative cases at each time point (day) between two groups. The reason it appears as if there are four animals in each group is because they were compared over four time points (four days). As indicated in the figure legend, the evaluation was performed using five animals in each group, and a representative picture is presented
5) Supp Fig. S6 was not included in my documents.
→Thank you for pointing out our mistake. The order was corrected.

Round 2
Reviewer 2 Report
In the legend of Fig. S6 (former S5), it should be 10 to the power of 4: "1.0 × 104 cells" instead of "1.0 × 104 cells" Otherwise, the points are addressed and I would support the acceptance of the manuscript.